# Development of a tool to assess beliefs about mythical causes of cancer: the Cancer Awareness Measure Mythical Causes Scale

Samuel G Smith,[1,2] Emma Beard,[2,3] Jennifer A McGowan,[2] Emma Fox,[4] Chloe Cook,[5] Radhika Pal,[2] Jo Waller,[2] Lion Shahab[2]

¹Leeds Institute of Health Sciences, University of Leeds, Leeds, UK
²Research Department of Behavioural Science and Health, University College London, London, UK
³Research Department of Clinical, Educational and Health Psychology, University College London, London, UK
⁴Division of General Internal Medicine, Northwestern University, Chicago, IL, USA
⁵Shift Design, London, UK

**Correspondence to**
Dr Samuel G Smith;
S.Smith1@leeds.ac.uk

## ABSTRACT

**Objectives** We aimed to develop a reliable and valid measure to assess public beliefs in mythical causes of cancer: the Cancer Awareness Measure–MYthical Causes Scale (CAM-MYCS).

**Design and setting** Cancer myth items were generated from a literature review, social media and interviews (n=16). The CAM-MYCS was prepared by reducing items using (a) an online sample (n=527) with exploratory factor analysis and (b) cancer experts with Delphi methodology (n=13). To assess test–retest reliability and sensitivity to change, students (n=91) completed the CAM-MYCS at baseline and 1 week after exposure to information on lifestyle-related cancer causes or control information. Construct validity was tested by comparing CAM-MYCS scores between cancer experts (n=25) and students (n=91). Factor structure and internal reliability were investigated in a national sample (n=1993).

**Results** Out of 42 items generated, 12 were retained based on factor loadings, prevalence of endorsement and expert consensus. CAM-MYCS scores improved (fewer myths endorsed) among students exposed to information on cancer causes compared with the control group (p<0.001) and showed high test–retest reliability (r=0.90, p<0.001). Cancer experts reported higher CAM-MYCS scores (fewer myths endorsed) than students (p<0.001). The factor structure of the CAM-MYCS was confirmed in the national sample and internal reliability was high (α=0.86). Inclusion of the CAM-MYCS alongside items assessing knowledge of actual cancer causes did not affect responses.

**Conclusions** The CAM-MYCS tool is a reliable and valid tool assessing beliefs in mythical causes of cancer, and it can be used alongside items assessing known causes of cancer.

## Strengths and limitations of this study

► This is the first study to develop a valid and reliable tool for assessing public beliefs in mythical causes of cancer—The Cancer Awareness Measure–MYthical Causes Scale (CAM-MYCS).

► Mythical beliefs is a novel construct that could influence cancer prevention initiatives, and the CAM-MYCS can be used to evaluate the success of cancer awareness campaigns.

► While the CAM-MYCS was developed using iterative mixed-methods, it is possible that it does not reflect all common beliefs in mythical causes of cancer held by the public.

► Future studies are needed to develop cancer site-specific versions and explore variation in mythical beliefs between countries.

## INTRODUCTION

A number of environmental causes of cancer have been identified including smoking, alcohol consumption, overweight, physical inactivity and poor diet.[1 2] An estimated 40% of cancer cases could be avoided through optimal adherence to lifestyle and environmental factors.[3] General population studies suggest awareness of environmental and lifestyle causes of cancer is mixed, which may undermine efforts to change behaviour at a population level.[4–9]

Recognition of prominent causes of cancer such as smoking and use of sunbeds is generally high.[5 6 10 11] However, other lifestyle factors such as alcohol consumption, overweight and low fruit and vegetable intake are poorly recognised. Awareness of the role of lifestyle factors in cancer is particularly poor among men, lower socioeconomic status groups, ethnic minorities and people with lower levels of education.[6 8–10 12 13] A survey of Dutch patients with urinary bladder cancer shows the likelihood of attributing a cancer diagnosis to lifestyle factors is low, even among patients with known risk factors such as smoking.[14]

In addition to poor recognition of established causes of cancer, a sizeable minority of the public continue to endorse mythical causes for which there is no scientific consensus for a causal effect for example,

powerlines, deodorant and stress.[9–11] The majority of this work has been done in the USA and the UK. It is important to investigate the public's causal beliefs about cancer as the way in which we think about disease risk factors can affect treatment decision making and prevention behaviour.[15–17] Understanding how common such mythical beliefs are among the general population can help to guide campaigns attempting to improve public understanding of cancer.

At present, there is no reliable and validated tool to assess beliefs in mythical causes of cancer. Current research employs unsystematic approaches when deciding which myths to include as distractor items in surveys, or uses open-ended assessments that measure recall rather than recognition.[10 11] The Cancer Awareness Measure (CAM)[18] and the Awareness and Beliefs about Cancer (ABC) measure[19] are the most frequently used validated assessments of known risk factor awareness, however they do not include items assessing awareness of mythical causes of cancer. Therefore only awareness of known risk factors is assessed in population surveys and public health campaign evaluations.[20] Assessing belief in mythical causes of cancer may provide a complementary perspective in which to study the effects of public understanding of cancer on treatment decision-making and lifestyle behaviours.

We aimed to (a) identify beliefs about mythical causes of cancer held within the general public and (b) develop a reliable and valid tool to measure belief in these mythical causes: the Cancer Awareness Measure –Mythical Causes Scale (CAM-MYCS). The purpose of the measure is to identify and report the prevalence of belief in mythical risks, that is, currently unsubstantiated risks for cancer; a factor that may be associated with health behaviour choices. This new measure will help to characterise the population and has the potential to lead to tailored interventions aimed at debunking mythical beliefs.

## MATERIALS AND METHODS
### Item generation
#### Systematic review
In May 2015, we searched for quantitative and qualitative articles reporting beliefs in mythical causes of cancer in general population samples (online supplementary appendix 1). Searches were run in Medline, EMBASE, PsycINFO and PsycEXTRA. English text, peer-reviewed studies were included if they reported at least one myth about cancer causes from a general population sample largely (>50%) unaffected by cancer. We sought reports of non-cancer populations to ensure the items generated were relevant to the general population, which is the intended group for the CAM-MYCS tool. Studies were excluded if they only reported knowledge about true causes of cancer and did not measure cancer myths, were not peer-reviewed (eg, commentaries, editorials) or were not written in the English language. A researcher (RP) retrieved all beliefs about mythical causes of cancer from the articles and included them in an item pool.

#### Semistructured interviews
A market research agency recruited 16 participants from the UK general population for semistructured interviews. People were approached using purposive sampling, balanced across key sociodemographic variables (age, gender, ethnicity and occupation). Participants who consented to take part in the study were invited to an interview at University College London or Queen Mary University of London. A topic guide was designed to explore beliefs about mythical cancer causes among the general public (online supplementary appendix 2). Each interview lasted approximately 30 min. Participants received £40 each for their time. Interviews were tape recorded and transcribed verbatim. A researcher trained in qualitative methods (RP) reviewed the interview data for beliefs about mythical cancer causes and added them to the item pool.

#### Social media
Online newspaper articles with 'cancer' in the title or text, or that were indexed as 'cancer' reported between March and June 2015 were extracted using a Lexis Nexis search. The search was restricted to four news agencies representing the full political spectrum and both broadsheet and tabloid newspapers (Independent, The Times, The Daily Mail, The Mirror). The BBC online news website was also included because it is widely accessed and is apolitical. The online comments associated with each article were extracted and were the focus of the data analysis.

Between July and August 2015, we also extracted tweets related to perceived causes of cancer from the social media site 'Twitter' (http://www.twitter.com). We used the search terms 'cancer' AND ['cause' OR 'prevent' OR 'treat'] in an open-access software tool.[21] Tweets had to be produced by users with no commercial affiliation. Passive replicated messages (ie, retweets) were excluded. A researcher (CC) recorded the frequency of beliefs about mythical cancer causes within the online newspaper comments and tweets using content analysis.[22] Due to the volume of beliefs reported, we only included those which were reported at least 10 times.

#### Patient and public involvement
To identify further factors from a patient perspective, we recruited four individuals on to a patient and public involvement (PPI) panel. The representatives were a mixture of cancer survivors and relatives of cancer survivors recruited from a charity. Four individuals attended a presentation about our work and provided suggestions which were accommodated within the item pool.

### Item refinement
#### Researcher revision
In December 2015, three researchers (SS, LS, JM) used the latest scientific evidence[1] to examine the scientific consensus for the associations between each item and cancer. Similar items were combined (eg, physical

trauma/sports trauma). Items were removed using the following criteria: (1) an inability to be tested for a relationship with cancer (eg, fate), (2) evidence of a causal route to cancer (eg, hygiene) and (3) the suggested relationship for the item was cancer prevention rather than cause (eg, eating avocados). This created a pool of 42 approved items.

## Delphi analysis

A Delphi analysis was undertaken with experts (n=13). Delphi analysis involves the anonymous collection of data from experts originating from a range of backgrounds, aimed at developing an unbiased consensus.[23] The panel included experts in oncology, public health, primary care and behavioural science; all with professional interests in cancer.

In the first round, experts were asked to individually list all beliefs about mythical cancer causes they were aware of. In round 2, the list from round 1 was combined with the previously generated item pool. The experts identified items they believed were commonly endorsed myths about cancer causes in the general population. They were then asked to provide a list of the top 10 items which they felt were most frequently endorsed by the public. In round 3, the experts viewed a list of the most commonly approved items from round 2 and indicated items they felt should be excluded. The experts also viewed a list of items that were previously suggested for removal and were asked to identify any they felt should be kept in the final questionnaire. This resulted in a final consensus of items.

## Online survey

The aim of the online survey was to produce data for an exploratory factor analysis. The survey was done in parallel with the Delphi analysis using the pool of 42 items. An online panel (n=500) was recruited through a research agency. Non-respondents and those with incomplete responses were removed (n=27), resulting in 473 useable respondents. Participants were asked to complete a questionnaire consisting of demographic information and the 42 items identified in the item generation stage. Eleven correct cancer causes from the CAM were also included.[18]

Respondents were asked, 'How much do you agree that each of these can increase a person's chance of developing cancer?' Responses to each item were dichotomised into 'correct' ('disagree' or 'strongly disagree') and 'incorrect' ('agree', 'strongly agree' or 'not sure') responses. The order of the items was randomised for each participant, such that incorrect and correct items were not shown separately.

Items underwent principal component analysis using varimax rotation in SPSS V.24. To reduce the number of items, we observed the item loadings as well as the frequency with which myths were endorsed. Our a priori criteria for excluding items were: (1) failure to load strongly onto a single factor (loadings <0.4) to ensure internal validity; (2) items for which more than 85% or fewer than 15% of the participants gave either a correct or incorrect response, to ensure sufficient variance in data and avoid ceiling effects[24] and (3) items recommended for exclusion by the Delphi analysis to ensure construct validity.

The final scoring of the CAM-MYCS was designed such that higher scores reflect superior knowledge about mythical cancer causes, that is, one point was allocated for each myth that was correctly identified (ie, 'strongly disagree' and 'disagree' responses). These scores were transformed to a score of 0–100 using the per cent of maximum possible method.[25]

## Item validation
### Sensitivity to change and known groups

We recruited 91 students from University College London studying non-medical subjects using an e-newsletter. Twenty-one responses were excluded because of missing data, leaving data from 70 respondents. Participation was incentivised with entry into a prize draw for a £25 voucher. In an online survey, participants answered the 12-item CAM-MYCS measure at baseline and at 1-week follow-up. After completing the baseline questionnaire, the sample was randomised 1:1 to either the intervention or control group. The intervention group was sent an online link to a brief educational intervention describing general information regarding cancer development, the link between cancer and lifestyle behaviours and commonly held myths about cancer causes (online supplementary appendix 3). The control group did not receive any intervention.

To assess sensitivity to change, total CAM-MYCS scores for the intervention and control groups were compared at follow-up using repeated measures analysis of variance (ANOVA). Test–retest reliability of the CAM-MYCS measure was assessed by calculating Pearson's correlation coefficient for baseline and 1-week follow-up CAM-MYCS scores in the control condition.

Cancer experts (oncology nurses, scientists, cancer charity workers) were recruited through professional networks (n=25). The experts were invited via email, which included a link to a survey containing basic background information and the CAM-MYCS items. To determine construct validity, total CAM-MYCS scores for the students and experts were compared using repeated measures ANOVA and independent t-tests.[24] Analyses were done in SPSS V.24.0.

## National survey

Data were from the Attitudes and Beliefs about Cancer-UK Survey, a nationally representative population-based cross-sectional survey in England (n=1993). The survey was done between January and March 2016. This survey creates sample points using the 2001 Census small-area statistics and the Postcode Address File (stratified by social grade and Government Office Region) for random location sampling. Quotas for age, gender, children in the home, and working status were set for each

location. Data were collected using computer-assisted face-to-face interviews in the respondents' homes.

Participants completed demographic information, the CAM risk factor measure[18] and the finalised CAM-MYCS measure. The CAM measure contains 11 known cancer causes and uses the same responses categories as the CAM-MYCS measure. Respondents were randomised to complete the CAM alone or the CAM and the CAM-MYCS measure on a 1:2 basis. A randomised design was used to investigate whether the CAM-MYCS measure could be used alongside the CAM, without affecting responses. Similar CAM scores in each group would indicate that the inclusion of the CAM-MYCS measure did not affect responses. Overall, 1352 respondents were randomised to the CAM-MYCS and CAM condition and 641 respondents to CAM questions alone. Participants were excluded if they did not respond to all CAM-MYCS or CAM questions or if they used the same response for all items. This resulted in a sample size of 1327 for CAM-MYCS and CAM and 640 for CAM alone. Those with missing data on CAM-MYCS or CAM were less likely to be of white ethnicity, but there were no other differences.

Refusal rates were calculated to assess acceptability of the measure. Other missing data were handled using case-wise (or 'full information') maximum likelihood estimation. Using data from people who completed both the CAM and CAM-MYCS measures, we undertook a confirmatory factor analysis. This was done using the 'Lavaan' package in R V.3.3.1.[26] The following fit statistics were computed: the Bayesian Information Criterion (BIC), the Goodness of Fit Index (GFI) and the Root Mean Square Error of Approximation (RMSEA). A two-factor model with CAM and CAM-MYCS items loading appropriately onto the respective factors would indicate construct validity. Statistical significance was set at $p < 0.05$ in all studies.

## RESULTS
### Item generation
A total of 999 studies (k) were identified in the systematic review. Duplicates were removed (k=987 remaining), and titles and abstracts were screened (k=55 remaining). Following full text screening, 16 studies remained and underwent quality assessment using the Newcastle-Ottawa Scale.[27] Fifty nine beliefs about mythical cancer causes were identified (figure 1). The qualitative interview study yielded 33 beliefs. A total of 33557 tweets and online comments were reviewed, of which 671 met inclusion criteria. From these, 93 beliefs about mythical cancer causes were identified. The PPI panel identified four beliefs about mythical cancer causes. After the review, interviews, PPI panel and social media analysis, 103 unique beliefs about mythical cancer causes were included in the item pool.

### Item refinement
Following our inclusion and exclusion criteria, we reduced the item pool to 42 beliefs about mythical causes of cancer (table 1). The Delphi analysis yielded no additional items. After three survey rounds, the expert group reached consensus on 13 items, which were recommended for inclusion in the measure.

Data from an online survey including the 42-item pool complemented the Delphi analysis. The study sample are described in table 2. In an exploratory principal component analysis, a one-component model was observed using both Eigenfactor and scree-plot based criteria. All except three items (exposure to parabens, using illegal drugs, exposure to chemtrails) loaded strongly onto this factor (loadings ≥0.40), and therefore further item selection was based on a priori criteria. No items were removed because of insufficient or excessive correct responses. Twenty-five items were removed because too few respondents endorsed the belief (ie, answered incorrectly). Of the remaining items, two were removed (eating food containing sugar, using energy-efficient lightbulbs) as these were not endorsed by expert consensus in the Delphi analysis. This resulted in a final list of 12 items to be included in the CAM-MYCS tool (table 1).

### Item validation
#### Sensitivity to change and known groups
There were no differences in baseline scores for non-medical students (intervention group: M=46.5, SD=26.8 vs control group: M=48.0, SD=25.5; t(68)=0.24, p=0.81). However, only non-medical students who received information on lifestyle-related cancer causes after baseline assessment improved their CAM-MYCS scores at follow-up compared with non-medical students in the control condition (F(1,68)=18.47; p<0.001), indicating the measure is sensitive to change (intervention group: M=62.0, SD=31.0 vs control group: M=41.4, SD=27.6; t(68)=2.94, p=0.005). Test–retest reliability was high for the control group completing the CAM-MYCS measure at baseline and 1-week follow-up (r=0.90, p<0.001).

The average CAM-MYCS scores for cancer experts was higher than the non-medical students at baseline (M=78.3, SD=24.4; vs M=47.3, SD=26.0; t(93)=5.22, p<0.001). This indicates the CAM-MYCS successfully distinguishes between groups known to have different levels of knowledge.

#### National survey
The group completing both the CAM-MYCS and CAM measures were similar to those completing the CAM measure only (table 2). The CAM-MYCS measure had a low refusal rate (3.9%), indicating acceptable length and content. Responses were normally distributed, with a skewness of 0.25 (SE=0.07) and a kurtosis of 0.19 (SE=0.13), suggesting it captures a range of knowledge within the population. Mean CAM scores

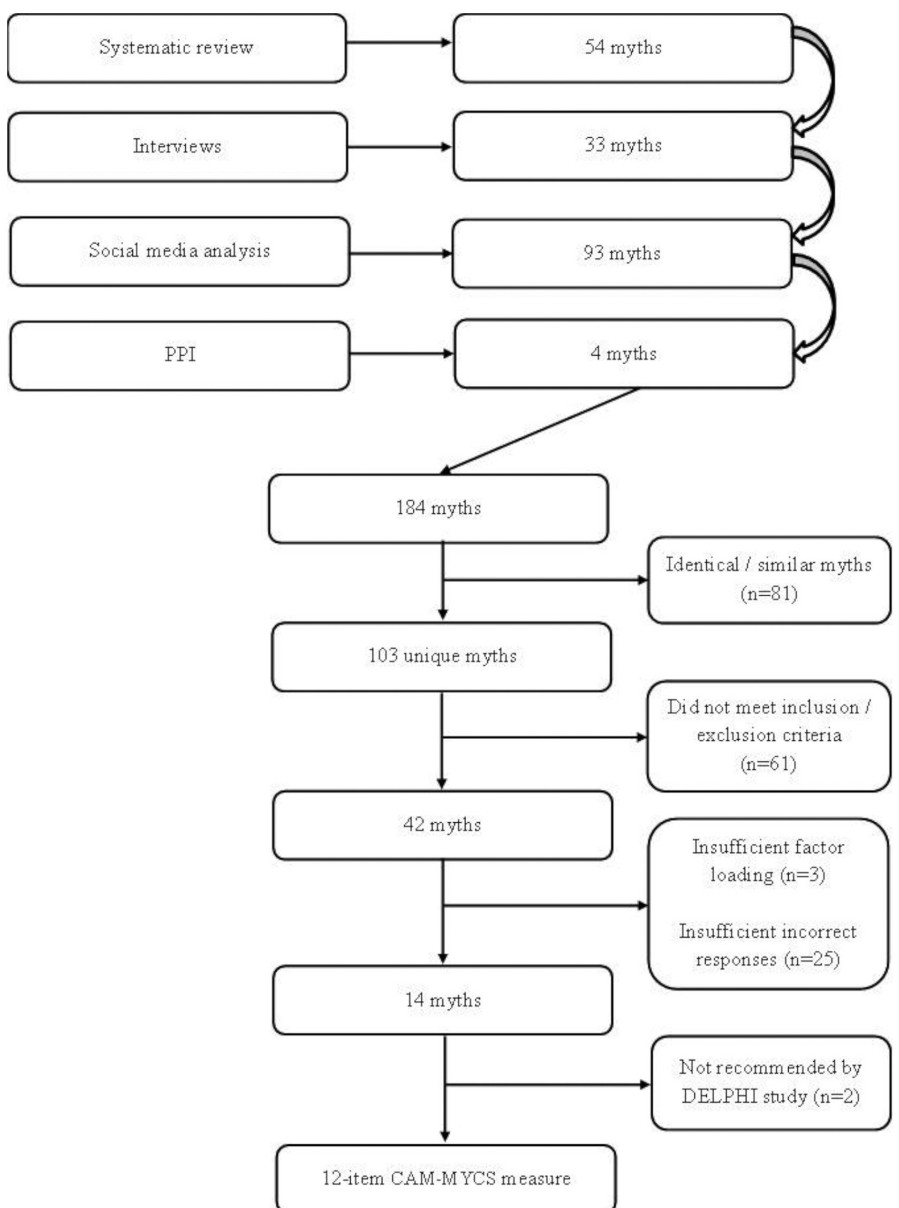

**Figure 1** Flow chart of CAM-MYCS development. CAM-MYCS, Cancer Awareness Measure–MYthical Causes Scale; PPI, patient and public involvement.

were comparable between the groups, demonstrating the inclusion of the CAM-MYCS items did not affect responses to the CAM (CAM+CAM-MYCS: M=52.78, SD=24.60 vs CAM only: M=52.32, SD=23.79, p=0.65).

Confirmatory factor analysis including both CAM and CAM-MYCS items suggested a two-factor solution provided a better fit than a one-factor model (difference $\chi^2(1)$=1302.6, p<0.001). Items belonging to the CAM and CAM-MYCS measures loaded onto the appropriate factors (table 3). The CAM-MYCS model was significantly improved following the removal of the item 'physical trauma' (difference $\chi^2(1)$=71.52, p<0.001), yielding good fit statistics (BIC=38 709.9, GFI=0.992, RMSEA=0.067, 95% CI 0.065 to 0.070). However, we decided to keep the item in the final measure because it was frequently identified within the item development phase, and good model fit was still

observed when it was included (BIC=42 450.6, GFI=0.992, RMSEA=0.054, 95% CI 0.052 to 0.056). The final 12-item CAM-MYCS measure had good internal reliability (Cronbach's α=0.86).

## DISCUSSION

In this iterative set of studies, we successfully identified commonly held beliefs about mythical causes of cancer and developed a valid and reliable measure to assess this construct. A range of perspectives were included to generate the items, including beliefs expressed on social media, in individual interviews and by a diverse set of experts. In a series of studies, 12 items emerged as providing optimal fit in both exploratory and confirmatory factor analyses. The inclusion of the CAM-MYCS

**Table 1** Items included in exploratory factor analysis for CAM-MYCS (n=473)

| Item | In final measure? | Recommended in Delphi? | Correct (%) | Unsure (%) | Incorrect (%) | Factor loading |
|---|---|---|---|---|---|---|
| Exposure to electromagnetic frequencies | Yes | Yes | 22.8 | 43.6 | 33.6 | 0.400 |
| Eating food containing additives | Yes | Yes | 32.1 | 37.5 | 30.4 | 0.551 |
| Living near power lines | Yes | Yes | 32.8 | 38.4 | 28.8 | 0.481 |
| Feeling stressed | Yes | Yes | 41.6 | 31.1 | 27.3 | 0.541 |
| Eating food containing artificial sweeteners | Yes | Yes | 38.9 | 35.9 | 25.2 | 0.577 |
| Using cleaning products | Yes | Yes | 40.6 | 34.9 | 24.5 | 0.588 |
| Eating genetically modified food | Yes | Yes | 40.8 | 34.9 | 24.3 | 0.523 |
| Using mobile phones | Yes | Yes | 39.1 | 36.8 | 24.1 | 0.571 |
| Using aerosol containers | Yes | Yes | 44.8 | 31.7 | 23.5 | 0.640 |
| Physical trauma, for example a punch or squeeze | Yes | Yes | 49.9 | 28.3 | 21.8 | 0.561 |
| Using microwave ovens | Yes | Yes | 52.6 | 28.8 | 18.6 | 0.617 |
| Drinking from plastic bottles | Yes | Yes | 56.2 | 26.7 | 17.1 | 0.650 |
| Exposure to parabens | No | Yes | 15.6 | 52.7 | 31.7 | 0.291 |
| Using illegal drugs | No | No | 22.6 | 34.9 | 42.5 | 0.379 |
| Exposure to chemtrails | No | No | 20.9 | 44.2 | 34.9 | 0.380 |
| Eating food containing sugar | No | No | 59.2 | 23.7 | 17.1 | 0.671 |
| Using energy-efficient light bulbs | No | No | 74.0 | 9.7 | 16.3 | 0.724 |
| Using cosmetics | No | No | 52.9 | 33.4 | 13.7 | 0.497 |
| Breast development as a teenager | No | No | 48.2 | 38.1 | 13.7 | 0.622 |
| Experiencing anger | No | No | 62.2 | 24.9 | 12.9 | 0.633 |
| Exposing cancer to the air | No | No | 61.5 | 25.6 | 12.9 | 0.546 |
| Experiencing depression | No | No | 57.9 | 29.4 | 12.7 | 0.611 |
| Using or being exposed to incense | No | No | 54.5 | 32.8 | 12.7 | 0.619 |
| Undergoing surgery | No | No | 59.4 | 28.3 | 12.3 | 0.591 |
| Exposure to WiFi signal | No | No | 57.5 | 30.9 | 11.6 | 0.660 |
| Using sunscreen | No | No | 69.6 | 19.0 | 11.4 | 0.557 |
| Eating dairy products | No | No | 68.5 | 20.9 | 10.6 | 0.675 |
| Using tampons | No | No | 65.1 | 24.8 | 10.1 | 0.596 |
| Using blood pressure medications | No | No | 57.5 | 34.3 | 8.2 | 0.668 |
| Using talcum powder | No | No | 54.1 | 37.7 | 8.2 | 0.610 |
| Frequent sexual activity with the same partner | No | No | 79.1 | 13.1 | 7.8 | 0.596 |
| Wearing tight clothing | No | No | 69.8 | 22.4 | 7.8 | 0.638 |
| Eating carbohydrates | No | No | 71.5 | 20.9 | 7.6 | 0.709 |
| Wearing an underwired bra | No | No | 71.0 | 21.4 | 7.6 | 0.710 |
| Carrying money in your bra | No | No | 77.4 | 15.4 | 7.2 | 0.674 |
| Receiving vaccinations | No | No | 69.8 | 23.0 | 7.2 | 0.714 |
| Consuming vitamin pills | No | No | 68.9 | 24.3 | 6.8 | 0.646 |
| Eating chewing gum | No | No | 73.6 | 20.1 | 6.3 | 0.648 |
| Experiencing jet lag | No | No | 74.8 | 19.1 | 6.1 | 0.727 |
| Eating food containing gluten | No | No | 69.3 | 24.6 | 6.1 | 0.745 |
| Exposure to the cold | No | No | 77.4 | 16.9 | 5.7 | 0.685 |
| Eating food containing soya | No | No | 71.0 | 23.3 | 5.7 | 0.722 |

Electromagnetic frequencies refers to non-ionising radiation of low and high frequencies such as WiFi and radio/TV frequencies.
CAM-MYCS, Cancer Awareness Measure – MYthical Causes Scale.

**Table 2** Participant characteristics for studies

| | Online panel study | Student and experts validation survey | | National survey | | |
| --- | --- | --- | --- | --- | --- | --- |
| | | Cancer experts (n=25) | Non-medical students (n=70) | Overall (n=1967) | CAM-MYCS and CAM (n=1327) | CAM alone (n=640) |
| | n=498 | | | | | |
| **Age** | | | | | | |
| Mean (SD) | 42.2 (15.5) | 38.2 (10.8) | 24.7 (7.3) | 43.7 (16.0) | 43.9 (15.9) | 43.4 (16.1) |
| ≤30 | 31.5 (156) | 32.0 (8) | 84.3 (59) | 27.6 (566) | 27.4 (363) | 28.7 (184) |
| 31–40 | 17.3 (86) | 36.0 (9) | 11.4 (8) | 18.5 (378) | 17.6 (233) | 18.9 (121) |
| 41–50 | 18.5 (92) | 12.0 (3) | 1.4 (1) | 16.2 (332) | 17.5 (232) | 13.9 (89) |
| 51–60 | 18.1 (90) | 16.0 (4) | 2.9 (2) | 17.5 (359) | 17.0 (225) | 19.4 (124) |
| 61+ | 14.5 (72) | 4.0 (1) | 0 (0) | 20.2 (412) | 20.6 (274) | 19.1 (122) |
| **Gender** | | | | | | |
| Male | 39.2 (195) | 8.0 (2) | 25.7 (18) | 46.8 (921) | 46.0 (610) | 48.6 (311) |
| Female | 60.6 (302) | 92.0 (23) | 74.3 (52) | 53.2 (1046) | 54.0 (717) | 51.4 (329) |
| Prefer not to say | 0.2 (1) | 0 (0) | 0 (0) | 0 (0) | 0 (0) | 0 (0) |
| **Ethnicity** | | | | | | |
| White British | 82.1 (409) | 76.0 (19) | 28.6 (20) | 75.6 (1548) | 76.1 (1010) | 75.9 (486) |
| White other | 5.4 (27) | 12.0 (3) | 28.6 (20) | 8.4 (173) | 8.2 (109) | 8.8 (56) |
| Other | 12.5 (62) | 12.0 (3) | 42.8 (30) | 16.0 (327) | 15.7 (2.08) | 15.3 (98) |
| **Education** | | | | | | |
| Degree or higher | 36.1 (180) | 80.0 (20) | 77.1 (54) | 26.0 (533) | 28.1 (373) | 23.0 (147) |
| Higher education | 9.0 (45) | 12.0 (3) | 8.6 (6) | 11.2 (229) | 11.6 (154) | 11.1 (71) |
| A-Level* | 21.1 (105) | 4.0 (1) | 14.3 (10) | 13.7 (281) | 14.1 (187) | 13.8 (88) |
| ONC/BTEC | 4.0 (20) | 0 (0) | 0 (0) | 6.1 (125) | 5.9 (78) | 7.0 (45) |
| GCSE/O-Level | 23.7 (118) | 0 (0) | 0 (0) | 25.0 (513) | 24.5 (325) | 25.8 (165) |
| None | 4.0 (20) | 0 (0) | 0 (0) | 13.8 (282) | 12.4 (164) | 15.0 (96) |
| Other | 0.8 (4) | 4.0 (1) | 0 (0) | 2.8 (58) | 2.5 (33) | 3.8 (24) |
| Prefer not to say | 1.2 (6) | 0 (0) | 0 (0) | 1.3 (27) | 1.0 (13) | 0.6 (4) |

Figures reported are % (n) for all apart from mean age (SD).

*For the student and expert validation survey, this category also included secondary education diplomas.

BTEC, Business and Technology Education Council; CAM, Cancer Awareness Measure; CAM-MYCS, Cancer Awareness Measure Mythical Causes Scale; GCSE, General Certificate of Secondary Education; ONC, Ordinary National Certificate.

items alongside the CAM assessment did not influence awareness of actual cancer causes. Both item sets loaded appropriately onto the hypothesised factors in confirmatory factor analysis. We therefore encourage the two assessments to be used alongside each other to provide a more accurate assessment of public knowledge about cancer risk.

Evaluations of public awareness campaigns rely on assessments that only include established lifestyle behaviours related to cancer development.[20] However, when such campaigns involve face-to-face interaction between healthcare professionals and the public, a portion of the conversations are likely to involve discussions about cancer myths. The CAM-MYCS can be embedded within these evaluations to examine if myths are being adequately addressed by such campaigns. Furthermore, these outcome data can inform the content of the written information disseminated within such campaigns. Until

now, population surveys investigating the prevalence of beliefs about mythical causes of cancer have used assessments that are not validated. Progress can now be made in reliably and accurately assessing public beliefs in mythical causes of cancer.

Separately, we have reported the prevalence of CAM-MYCS items, their sociodemographic correlates and tested for associations with cancer prevention behaviours.[28] Briefly, participants showed poor awareness of factors not causally linked with cancer, with only a third of mythical cancer causes identified correctly. The most commonly endorsed cancer myths were stress (41.7%), food additives (41.1%), exposure to non-ionising electromagnetic frequencies (34.7%) and genetically modified foods (34.1%). Perhaps counterintuitively, better awareness of mythical risk factors was associated with a greater likelihood of smoking and having a higher aggregated behaviour risk score (composed of smoking, physical

**Table 3** Unstandardised and standardised loadings for confirmatory models (n=1327)

| | Single factor model | | | Two-factor model | | | | | |
| | Factor 1 | | | Factor 1 (CAM) | | | Factor 2 (CAM-MYCS) | | |
| | Unstandardised | | Standardised | Unstandardised | | Standardised | Unstandardised | | Standardised |
| | Estimate | SE | Estimate | Estimate | SE | Estimate | Estimate | SE | Estimate |
|---|---|---|---|---|---|---|---|---|---|
| CAM 1 | 1 | 0 | 0.506 | 1 | 0 | 0.543 | | | |
| CAM 2 | 1.098 | 0.055 | 0.584 | 1.067 | 0.05 | 0.608 | | | |
| CAM 3 | 1.2 | 0.057 | 0.651 | 1.204 | 0.052 | 0.7 | | | |
| CAM 4 | 1.156 | 0.056 | 0.625 | 1.096 | 0.051 | 0.636 | | | |
| CAM 5 | 1.063 | 0.055 | 0.553 | 1.077 | 0.051 | 0.6 | | | |
| CAM 6 | 1.046 | 0.049 | 0.662 | 1.076 | 0.046 | 0.73 | | | |
| CAM 7 | 1.165 | 0.057 | 0.615 | 1.177 | 0.052 | 0.667 | | | |
| CAM 8 | 1.247 | 0.059 | 0.665 | 1.148 | 0.052 | 0.656 | | | |
| CAM 9 | 1.179 | 0.057 | 0.638 | 1.109 | 0.051 | 0.644 | | | |
| CAM 10 | 1.142 | 0.054 | 0.667 | 1.162 | 0.05 | 0.727 | | | |
| CAM 11 | 1.297 | 0.06 | 0.677 | 1.274 | 0.055 | 0.712 | | | |
| CAM-MYCS 1 | 1.177 | 0.06 | 0.669 | | | | 1 | 0 | 0.69 |
| CAM-MYCS 2 | 1.356 | 0.065 | 0.742 | | | | 1.12 | 0.044 | 0.746 |
| CAM-MYCS 3 | 1.207 | 0.062 | 0.665 | | | | 1.056 | 0.043 | 0.705 |
| CAM-MYCS 4 | 1.253 | 0.066 | 0.644 | | | | 1.004 | 0.047 | 0.626 |
| CAM-MYCS 5 | 1.278 | 0.063 | 0.703 | | | | 1.074 | 0.044 | 0.717 |
| CAM-MYCS 6 | 1.127 | 0.059 | 0.647 | | | | 0.937 | 0.042 | 0.652 |
| CAM-MYCS 7 | 1.165 | 0.061 | 0.653 | | | | 1.043 | 0.043 | 0.708 |
| CAM-MYCS 8 | 1.259 | 0.064 | 0.676 | | | | 1.09 | 0.045 | 0.71 |
| CAM-MYCS 9 | 1.208 | 0.06 | 0.693 | | | | 1.022 | 0.042 | 0.713 |
| CAM-MYCS 10 | 0.967 | 0.059 | 0.526 | | | | 0.811 | 0.044 | 0.532 |
| CAM-MYCS 11 | 1.168 | 0.061 | 0.65 | | | | 1.046 | 0.043 | 0.706 |
| CAM-MYCS 12 | 1.13 | 0.059 | 0.653 | | | | 0.971 | 0.042 | 0.68 |

For the two-factor model the covariance between the two factors was 0.352 (p<0.001).

CAM, Cancer Awareness Measure; CAM-MYCS, Cancer Awareness Measure Mythical Causes Scale.

activity, overweight, fruit and vegetables and alcohol consumption).

Understanding the extent to which mythical beliefs improve or undermine attempts to change health behaviours could inform the development of cancer prevention interventions and public health strategies. This measure also has implications for assessing the beliefs of people diagnosed with cancer. Failure to attribute cancer to known risk factors is a recognised phenomenon,[18 29] and the extent to which people associate mythical factors with their own diagnosis is not known. Such studies can now be reliably undertaken and have the potential to inform patient and provider dialogue.

There are limitations with the CAM-MYCS tool. Given the limitations of empiricism and the impossibility to prove a negative, we are unable to completely rule out causal relationships between the factors included within the tool and the development of cancer. The items were carefully chosen on the basis of scientific consensus, using reports from leading agencies,[1] and experts from a range of relevant disciplines within the Delphi study. It is possible that future research investigating the effects of these factors on carcinogenesis will report findings to convince the scientific community of a likely causal relationship. For instance, there is some preliminary evidence of a weak association between certain forms of cancer and mobile phone use[30] and non-ionising electromagnetic radiation more generally.[31] However, the extent to which this is causal is still debated.[32] If scientific consensus changes, the CAM-MYCS should be adapted to reflect the latest evidence. It is also important for users of the tool to note that we are referring to non-ionising electromagnetic frequencies within the scale; and this should be distinguished from other forms of radiation known to cause cancer, such as ultraviolet radiation.

The studies involved in the development of the tool also have limitations. While a range of different approaches were used to develop the item pool, each source may contain a biased sample. It is therefore possible that the CAM-MYCS measure does not reflect all common beliefs

in mythical causes of cancer held by the public. Beliefs about risk factors may vary by cancer site[6 33 34]; this may not be captured by the CAM-MYCS, and site-specific versions could be developed in the future. The online sample had a higher prevalence of young, female, White British and highly educated individuals than would be expected. This may affect generalisability of these findings. Mythical beliefs may emerge over time, and therefore a revision of the CAM-MYCS may be needed in the future. Finally, this work was done in UK samples. The prevalence of beliefs in mythical cancer causes may vary across countries. Such beliefs are likely to be influenced strongly by culture and social environments, and international variation should be tested in future analyses.[35 36]

## CONCLUSIONS

We used a series of iterative studies to demonstrate the reliability and validity of the CAM-MYCS measure in assessing beliefs in mythical causes of cancer among the general public. Of importance to public awareness campaigns and the evaluation of interventions, the CAM-MYCS measure can be used alongside the CAM without concerns that CAM responses will be affected. This approach is recommended as it is likely to provide a more accurate assessment of public knowledge about cancer aetiology than current strategies.

**Acknowledgements**  We would like to thank the ABACUS study team for their support in conducting the survey.

**Contributors**  SGS, LS: conceived the original idea for this study and obtained funding. JAM, SGS, LS: managed the day-to-day running of the study. LS, SGS, JAM, CC, EF, RP: were involved in data collection (including the systematic review, interviews, social media analysis, and the student, online and national surveys). LS, EB, JAM: undertook the data analysis. SS: wrote the initial draft with further input from EB, JAM, EF, CC, RP, JW, LS. LS: is guarantor for this article. SGS, EB, JAM, EF, CC, RP, JW, LS: reviewed and approved the final version. All researchers listed as authors are independent from the funders, and all final decisions about the research were taken without constraint by the investigators. LS, EB, JAM: had full access to all the data in the study. SGS, LS: had final responsibility for the decision to submit for publication.

**Funding**  This work was supported by a Cancer Research UK/Bupa Foundation Innovation Award (C42785/A20811). SS and JW are supported by Cancer Research UK Fellowships (C42785/A17965 and C7492/A17219 respectively). LS is a member of the UK Centre for Tobacco and Alcohol Studies (UKCTAS), funded under the auspices of the above UK Clinical Research Collaboration (MR/K023195/1).

**Disclaimer**  The funders had no role in the collection, analysis or interpretation of data; in the writing of the report or in the decision to submit the article for publication.

**Competing interests**  LS has received honoraria for talks, an unrestricted research grant and travel expenses to attend meetings and workshops by pharmaceutical companies that make smoking cessation products (Pfizer, Johnson&Johnson) and has acted as paid reviewer for grant awarding bodies and as a paid consultant for health care companies. Other research has been funded by the government, a community-interested company (National Centre for Smoking Cessation) and charitable sources. SS is an academic advisor for LUTO.

**Patient consent**  Obtained.

**Ethics approval**  University College London ethical approval was gained for all studies.

**Provenance and peer review**  Not commissioned; externally peer reviewed.

**Data sharing statement**  Anonymised data will be shared upon request after all planned manuscripts have been published by the authors. A data sharing agreement will be generated, and all data will be anonymised prior to sharing.

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
