## [Reviewer comments · BMJ Open]

ARTICLE DETAILS

TITLE (PROVISIONAL)	The development of a tool to assess beliefs about mythical causes of cancer: the Cancer Awareness Measure Mythical Causes Scale
AUTHORS	Smith, Sam; Beard, Emma; McGowan, Jennifer; Fox, Emma; Cook, Chloe; Pal, Radhika; Waller, Jo; Shahab, Lion

VERSION 1 – REVIEW

REVIEWER	Pror.(Dr.) Bhudev C. Das Amity Institute of Molecular Medicine & Stem Cell Research (AIMMSCR), Amity University Uttar Pradesh, Amity Campus, Sector - 125, Noida INDIA
REVIEW RETURNED	26-Apr-2018

GENERAL COMMENTS	I have gone through the manuscript entitled “The development of a tool to assess beliefs about mythical causes of cancer: the cancer awareness measure mythical causes scale”, submitted by Samuel G Smith et al, for its publication in BMJ Open. The manuscript suffers from several serious drawbacks and my specific comments are as follows: 1. The model only tests the level of awareness of the risk factors related to cancer (CAM) and the knowledge of the mythical beliefs associated with the development. I, however, do not see potential for its use in cancer prevention, unless there exists particular practices at the individual level, associated with the myths that exacerbates the symptoms or it is an established risk factor for cancer. Did the researchers identify any such practice related to the myths? The big question here remains as to what the CAM-MYCS model tries to achieve and why is it required along with CAM? Justification is required as to how CAM-MYCs will be better to evaluate success of cancer awareness campaigns? How does identification of mythical beliefs in general public with regard to cancer improve cancer prevention/survival?2. Table 1 - Please categorize the table into CAM & CAM-MYCS.3. I am not sure if the list of final 12 mythical items added to CAM-MYCS are not a part of any study showing a causal/associative relationship and/or are currently being studied for a link with cancer.4. The discussion section needs to be elaborated, mentioning the mythical beliefs that were more prominently endorsed by the general public during the survey (both online panel study & national survey).5. Does the leaflet introduced for the intervention group include or mention the typical myths associated with cancer or just talks about the correct/established factors related to cancer? Because if the
---

	leaflets is deprived of this information then how come the intervention group got higher scores during follow up? 6. Ethical issues related to incentivization of participant during or after recruitment, does it lead to voluntary involvement of participants? It is possible that the respondents participated in the study only for money. This may have biased participant recruitment. 7. Result section – National survey : Mean CAM score comparable b/w both groups (CAM only & CAM-MYCS) & Discussion section – Inclusion of CAM-MYCS did not influence awareness of actual cancer causes.(This implies that the knowledge related to the established factors for cancer is equal in both groups, indicating good knowledge because even after introducing the MYCS item, the respondents could identify/not identify CAM factors appropriately). However, it would be interesting to see the mean CAM & CAM-MYCS scores among the second group that answered both CAM & CAM-MYCS in the national survey and how well they performed on CAM-MYCS. Need to apply statistical interpretation. 8. The conclusion is over ambitious one which is not supported or validated by the results presented. Accordingly, abstract needs modification. In summary, the manuscript needs major revision and answers to several serious questions that have been raised above. The acceptability of this manuscript depends on the extent the authors will be able to clarify the points. Manuscript be revised and resubmitted for reconsideration. I will be happy to review the manuscript again.
--	---

REVIEWER	Qing Liu Amgen Inc. United States
REVIEW RETURNED	01-May-2018

GENERAL COMMENTS	This is a well-written manuscript. The authors conducted intensive analyses to evaluate the reliability and validity of the CAM-MYCS tool in assessment of public beliefs in mythical causes of cancer. One minor comment that I hope could help the authors improve the paper. On page 11, data analysis was conducted based on complete data. To justify this method, it is good to describe the missing pattern and/or describe the subjects' characteristics whose CAM and/or CAM-MYCS measures were incomplete.
---

REVIEWER	Jada Hamilton, PhD, MPH Assistant Attending Psychologist Memorial Sloan Kettering Cancer Center USA
REVIEW RETURNED	21-Jun-2018

GENERAL COMMENTS	This manuscript describes a systematic, multistage approach to the creation of a reliable and valid measure of public beliefs about mythical causes of cancer. Individuals' beliefs about inaccurate, mythical causes of cancer may interfere with their responses to public health communication campaigns aimed at cancer prevention and control efforts, thus the creation of a measure assessing this construct is a worthwhile contribution. This manuscript benefits from a thoughtful development and validation approach utilizing a variety of relevant data sources (experts, public discourse via news websites and Twitter, students, UK national sample). The
---

	manuscript is clear and well written. However, there are several issues that need to be addressed. Specific concerns, listed in order of appearance in the manuscript, include:  1) Abstract: For clarity, in the Results section of the Abstract please clarify that “improved” and “higher” scores indicate less endorsement of mythical beliefs. 2) Introduction, Page 4, lines 45-47: Can more be said regarding the decision to highlight the CAM and ABC? For example, are these the most commonly used measures, or the only validated measures? 3) Materials & Methods, Page 7, line 47: Please add information about when the reviews of latest evidence and literature searches to evaluate the veracity of the items/causes were conducted. 4) Materials & Methods, Page 11, lines 17-19: Given text on page 14, should the GFI also be listed as an evaluated fit statistic? 5) Results, Page 12, Lines 7-9: Please provide additional information about the article screening and quality assessment processes. For example, were titles and abstracts evaluated first, and then only potentially relevant full-text articles reviewed? How was quality assessed? 6) Results, Page 14, Lines 5 and 9: To further improve clarity, please include the confidence intervals for the RMSEA values. 7) Discussion, Page 15, line 41: Please consider providing a brief summary of the main findings of reference #26. The inclusion of such details may help to bolster the significance of measuring the construct of mythical beliefs about cancer. 8) Table 1: Please check to be sure that the footnote applies to this table (it does not appear that any text is italicized). 9) Table 2: In the column for “online panel study”, is there a typo in the row for “Age, Mean (SD)” (i.e., SD of 88.4 years)? 10) Supplementary Information, Cancer Information Leaflet: Content under the sections on sunlight and diet is cut off.
--	--

VERSION 1 – AUTHOR RESPONSE

Reviewer: 1

Reviewer Name: Pror.(Dr.) Bhudev C. Das

I have gone through the manuscript entitled “The development of a tool to assess beliefs about mythical causes of cancer: the cancer awareness measure mythical causes scale”, submitted by Samuel G Smith et al, for its publication in BMJ Open. The manuscript suffers from several serious drawbacks and my specific comments are as follows:

Author reply: Thank you for reviewing our manuscript. Please find our responses to your comments below.

1. The model only tests the level of awareness of the risk factors related to cancer (CAM) and the knowledge of the mythical beliefs associated with the development. I, however, do not see potential for its use in cancer prevention, unless there exists particular practices at the individual level, associated with the myths that exacerbates the symptoms or it is an established risk factor for cancer. Did the researchers identify any such practice related to the myths? The big question here remains as to what the CAM-MYCS model tries to achieve and why is it required along with CAM? Justification is required as to how CAM-MYCS will be better to evaluate success of cancer awareness campaigns? How does identification of mythical beliefs in general public with regard to cancer improve cancer prevention/survival?

Author reply: The purpose of developing the CAM-MYCS scale was to establish if a valid and reliable tool assessing public knowledge of cancer myths was associated with behaviours related to cancer prevention (e.g. smoking, physical activity, overweight). We have published data within a special edition of the European Journal of Cancer investigating these associations, and therefore we have not

repeated them here (see Shahab et al., (2018) Prevalence of beliefs about actual and mythical causes of cancer and their association with socio-demographic and health-related characteristics: Findings from a cross-sectional survey in England. *European Journal of Cancer*). We have however briefly described the findings of this paper in the discussion. The aim of the current manuscript is to transparently report the development process for the scale.

Description of Shahab et al., 2018: 'Briefly, participants showed poor awareness of factors not causally linked with cancer, with only a third of mythical cancer causes identified correctly. The most commonly endorsed cancer myths were stress (41.7%), food additives (41.1%), exposure to non-ionizing electromagnetic frequencies (34.7%), and genetically modified foods (34.1%). Perhaps counterintuitively, better awareness of mythical risk factors was associated with a greater likelihood of smoking and having a higher aggregated behaviour risk score (composed of smoking, physical activity, overweight, fruit and vegetables and alcohol consumption).' (Discussion, page 15-16)

We have provided additional description regarding the aim of the CAM-MYCS scale and why it is needed alongside measures such as the CAM:

'It is important to investigate the public's causal beliefs about cancer as the way in which we think about disease risk factors can affect treatment decision-making and prevention behaviour.[15–17] Understanding how common such mythical beliefs are among the general population can help to guide campaigns attempting to improve public understanding of cancer.' (Introduction, page 4)

'The Cancer Awareness Measure (CAM)[18] and the Awareness and Beliefs about Cancer (ABC) measure[19] are the most frequently used validated assessments of known risk factor awareness, however they do not include items assessing awareness of mythical causes of cancer. Therefore only awareness of known risk factors is assessed in population surveys and public health campaign evaluations [20]. Assessing belief in mythical causes of cancer may provide a complementary perspective in which to study the effects of public understanding of cancer on treatment decision-making and lifestyle behaviours.' (Introduction, page 4-5)

'Evaluations of public awareness campaigns rely on assessments that only include established lifestyle behaviours related to cancer development.[20] However, when such campaigns involve face-to-face interaction between healthcare professionals and the public, a portion of the conversations are likely to involve discussions about cancer myths. The CAM-MYCS can be embedded within these evaluations to examine if myths are being adequately addressed by such campaigns.' (Discussion, page 15)

2. Table 1 - Please categorize the table into CAM & CAM-MYCS.

Author reply: We think you may have misinterpreted this table; Table 1 only includes items from the CAM-MYCS tool. We have altered the title to make this clearer.

3. I am not sure if the list of final 12 mythical items added to CAM-MYCS are not a part of any study showing a causal/associative relationship and/or are currently being studied for a link with cancer.

Author reply: Thank you for raising this important issue. Many of these factors have been investigated for a potentially causal link with the development of cancer, and we expect this research to be ongoing. As such, we cannot rule out the possibility that the items included in the CAM-MYCS scale may in the future be causally linked with cancer. To address this comment, we have included the following text:

'There are limitations with the CAM-MYCS tool. Given the limitations of empiricism and the impossibility to prove a negative, we are unable to completely rule out causal relationships between the factors included within the tool and the development of cancer. The items were carefully chosen on the basis of scientific consensus, using reports from leading agencies [1], and experts from a range of relevant disciplines within the Delphi study. It is possible that

future research investigating the effects of these factors on carcinogenesis will report findings to convince the scientific community of a likely causal relationship. For instance, there is some preliminary evidence of a weak association between certain forms of cancer and mobile phone use [30] and non-ionizing electromagnetic radiation more generally [31]. However, the extent to which this is causal is still debated [32]. If scientific consensus changes, the CAM-MYCS scale should be adapted to reflect the latest evidence. It is also important for users of the tool to note that we are referring to non-ionizing electromagnetic frequencies within the scale; and this should be distinguished from other forms of radiation known to cause cancer, such as ultraviolet radiation.’ (Discussion, Page 16)

We have also slightly altered the way in which we described the cancer myths throughout the text to reflect a more cautious approach.

4. The discussion section needs to be elaborated, mentioning the mythical beliefs that were more prominently endorsed by the general public during the survey (both online panel study & national survey).

Author reply: Thank you for this suggestion. The data you are referring to are reported in the separate manuscript cited above (Shahab et al., 2018 European Journal of Cancer). As provided above, we have included a brief description of our findings within the discussion.

5. Does the leaflet introduced for the intervention group include or mention the typical myths associated with cancer or just talks about the correct/established factors related to cancer? Because if the leaflet is deprived of this information then how come the intervention group got higher scores during follow up?

Author reply: Thank you for this question. As described in the methods, the intervention leaflet contained information on some of the commonly-held myths about cancer causes. This control group did not receive this information, and this explains why the intervention group were more aware of cancer myths. The intervention leaflet is included in Appendix 3.

6. Ethical issues related to incentivization of participant during or after recruitment, does it lead to voluntary involvement of participants? It is possible that the respondents participated in the study only for money. This may have biased participant recruitment.

Author reply: Thank you for raising this issue, however we do not believe this is problematic. Incentivising participation through payment is common practice, and can address some bias created by the recruitment of only highly intrinsically motivated individuals. We did take some steps to ensure all responses in the online survey were valid, such as by removing individuals who provided responses that were obviously provided only for monetary gain (e.g. very quick responses with the same answer for each item).

7. Result section – National survey: Mean CAM score comparable b/w both groups (CAM only & CAM-MYCS) & Discussion section – Inclusion of CAM-MYCS did not influence awareness of actual cancer causes. (This implies that the knowledge related to the established factors for cancer is equal in both groups, indicating good knowledge because even after introducing the MYCS item, the respondents could identify/not identify CAM factors appropriately). However, it would be interesting to see the mean CAM & CAM-MYCS scores among the second group that answered both CAM & CAM-MYCS in the national survey and how well they performed on CAM-MYCS. Need to apply statistical interpretation.

Author reply: Thank you for this suggestion. As per our response to comment 4, data on the prevalence of the beliefs within the CAM-MYCS tool are found in the paper by Shahab et al., (2018). We have provided some key findings of that study in the discussion.

8. The conclusion is over ambitious one which is not supported or validated by the results presented. Accordingly, abstract needs modification.

Author reply: It is unclear to us what aspect of the conclusion is over ambitious. Upon reading the conclusion again, we believe all of the statements included are supported by the data i.e. the CAM-

MYCS is 1) valid and reliable, 2) it can be used alongside the CAM and 3) we recommend this approach as it provides a more accurate assessment of public knowledge about cancer than current approaches (CAM alone). We also do not believe any changes need to be made to the abstract conclusion.

In summary, the manuscript needs major revision and answers to several serious questions that have been raised above. The acceptability of this manuscript depends on the extent the authors will be able to clarify the points. Manuscript be revised and resubmitted for reconsideration. I will be happy to review the manuscript again.

Author reply: Thank you for reviewing our manuscript. We have addressed the concerns raised where appropriate, and have provided rebuttals where we disagree.

Reviewer: 2

Reviewer Name: Qing Liu

This is a well-written manuscript. The authors conducted intensive analyses to evaluate the reliability and validity of the CAM-MYCS tool in assessment of public beliefs in mythical causes of cancer. One minor comment that I hope could help the authors improve the paper. On page 11, data analysis was conducted based on complete data. To justify this method, it is good to describe the missing pattern and/or describe the subjects' characteristics whose CAM and/or CAM-MYCS measures were incomplete.

Author reply: Thank you for this suggestion. We have now described the pattern of missing data:

'Those with missing data on CAM-MYCS or CAM were less likely to be of white ethnicity but there were no other differences.' (Method, page 11).

Reviewer: 3

Reviewer Name: Jada Hamilton, PhD, MPH

This manuscript describes a systematic, multistage approach to the creation of a reliable and valid measure of public beliefs about mythical causes of cancer. Individuals' beliefs about inaccurate, mythical causes of cancer may interfere with their responses to public health communication campaigns aimed at cancer prevention and control efforts, thus the creation of a measure assessing this construct is a worthwhile contribution. This manuscript benefits from a thoughtful development and validation approach utilizing a variety of relevant data sources (experts, public discourse via news websites and Twitter, students, UK national sample). The manuscript is clear and well written. However, there are several issues that need to be addressed.

Author reply: Thank you for reviewing our manuscript, and for your helpful comments.

Specific concerns, listed in order of appearance in the manuscript, include:

- 1) Abstract: For clarity, in the Results section of the Abstract please clarify that "improved" and "higher" scores indicate less endorsement of mythical beliefs.

Author reply: Thank you for this helpful suggestion. We have now implemented it within the abstract.

- 2) Introduction, Page 4, lines 45-47: Can more be said regarding the decision to highlight the CAM and ABC? For example, are these the most commonly used measures, or the only validated measures?

Author reply: Thank you for this suggestion. These measures are the most frequently used validated assessments. This information has now been included within the text.

- 3) Materials & Methods, Page 7, line 47: Please add information about when the reviews of latest evidence and literature searches to evaluate the veracity of the items/causes were conducted.

Author reply: These reviews were undertaken in December 2015. This information has now been included within the text.

- 4) Materials & Methods, Page 11, lines 17-19: Given text on page 14, should the GFI also be listed as an evaluated fit statistic?

Author reply: Thank you for pointing this out. We have now included the GFI in the section describing the analysis of the national survey.

- 5) Results, Page 12, Lines 7-9: Please provide additional information about the article screening and quality assessment processes. For example, were titles and abstracts evaluated first, and then only potentially relevant full-text articles reviewed? How was quality assessed?

Author reply: Thank you for highlighting this. We have now provided additional detail describing the process:

'A total of 999 studies (k) were identified in the systematic review. Duplicates were removed (k=987 remaining), and titles and abstracts were screened (k=55 remaining). Following full text screening, 16 studies remained and underwent quality assessment using the Newcastle-Ottawa Scale [27].' (Results, page 12)

- 6) Results, Page 14, Lines 5 and 9: To further improve clarity, please include the confidence intervals for the RMSEA values.

Author reply: Thank you. The 95% confidence intervals for the RMSEA values have now been added.

- 7) Discussion, Page 15, line 41: Please consider providing a brief summary of the main findings of reference #26. The inclusion of such details may help to bolster the significance of measuring the construct of mythical beliefs about cancer.

Author reply: Thank you for this suggestion. We have now provided the following description of the findings from Shahab et al., 2018:

'Briefly, participants showed poor awareness of factors not causally linked with cancer, with only a third of mythical cancer causes identified correctly. The most commonly endorsed cancer myths were stress (41.7%), food additives (41.1%), exposure to non-ionizing electromagnetic frequencies (34.7%), and genetically modified foods (34.1%). Perhaps counterintuitively, better awareness of mythical risk factors was associated with a greater likelihood of smoking and having a higher aggregated behaviour risk score (composed of smoking, physical activity, overweight, fruit and vegetables and alcohol consumption).'
(Discussion, page 15-16)

- 8) Table 1: Please check to be sure that the footnote applies to this table (it does not appear that any text is italicized).

Author reply: Thank you, this text has now been removed.

- 9) Table 2: In the column for "online panel study", is there a typo in the row for "Age, Mean (SD)" (i.e., SD of 88.4 years)?

Author reply: Thank you for this observation, which has now been corrected.

- 10) Supplementary Information, Cancer Information Leaflet: Content under the sections on sunlight and diet is cut off.

Author reply: Thank you. We will check this with the article processing team if the manuscript is accepted. It is not cut-off on the version we submitted, so we may have to send this separately.

VERSION 2 – REVIEW

REVIEWER	Qing Liu Amgen Inc. USA
REVIEW RETURNED	21-Aug-2018

GENERAL COMMENTS	This is a well-written manuscript. Thank you for the response to my comments.
---

REVIEWER	Jada Hamilton, Ph.D., M.P.H. Memorial Sloan Kettering Cancer Center US
REVIEW RETURNED	04-Sep-2018

GENERAL COMMENTS	The authors have been responsive to all the reviewer critiques. These modifications have substantially improved the overall quality and clarity of this manuscript, and the work of the authors to make these changes is much appreciated. I have no remaining concerns.
--